# Prevalence and correlates of physical fighting among adolescents in Paraguay: Findings from the 2017 national school-based health survey

**Hiroko Taniguchi**[1]*, **Masood Ali Shaikh**[2]

**1** Department of Global Health Policy, School of International Health, Graduate School of Medicine, The University of Tokyo, Tokyo, Japan, **2** Independent Consultant, Karachi, Pakistan

* taniguchih@m.u-tokyo.ac.jp

## Abstract

**Data Availability Statement:** All data are publicly available on the WHO website: Paraguay - Global School-Based Student Health Survey 2017.

### Background

Interpersonal violence among adolescents is a serious public health issue across the globe and has been one of the leading causes of death among Paraguayan adolescents. This study aims to investigate the prevalence of physical fighting among adolescents in Paraguay in order to identify problematic fighting behaviour. We also aim to examine the correlates of physical fighting and the extent to which previously identified factors correlate with physical fighting.

### Methods

We used the Paraguay 2017 Global School-based Student Health Survey (GSHS). This survey collects health-related information on school-attending adolescents aged 13–17 years. We defined physical fighting as having participated in at least two physical fights in the previous 12 months. We chose 16 independent variables: 12 individual-level variables and four social-level variables. Multivariable logistic regression models were developed to identify factors associated with physical fighting. One of the limitations of this study is that it only captured the responses of the students who attended school on the day of the survey.

### Findings

A total of 3,149 students completed the survey questionnaire, with the response rates for the school, student, and total response being 100%, 87%, and 87%, respectively. In 2017, 8% of the survey participants (11.4% of the males, and 4.7% of the females) had been involved in two or more physical fights during the past 12 months. In the multivariable model, having been physically attacked, male gender, physical activity, alcohol use, early sexual debut, and suicide planning were significantly associated with involvement in physical fighting. Having helpful peers and supportive parents was not statistically significant in the model adjusted for all variables.

(https://extranet.who.int/ncdsmicrodata/index.php/catalog/651).

**Funding:** The authors received no specific funding for this work.

**Competing interests:** The authors have declared that no competing interests exist.

## Conclusions

Although Paraguay shows relatively lower prevalence of physical fighting than other countries, the high association between physical fighting and having been physically attacked is noteworthy. Considering the serious interpersonal violence among Paraguayan adolescents, preventive attributes should be considered, and further assessment of other types of interpersonal violence should be made.

## Introduction

Youth violence refers to interpersonal violence which occurs among individuals aged 10–29 years; the violence includes a range of acts from bullying, emotional abuse, and physical fighting, to more severe sexual and physical assault, and homicide [1,2]. Interpersonal violence among adolescents is a serious public health issue across the globe and, in particular, for boys aged 15–19 years; it is the second leading cause of deaths in low- and middle-income countries (LMIC) and the third leading cause in high-income countries (HIC) [3,4]. Interpersonal violence accounts for 11% of the total number of deaths of adolescent boys in both income settings [4].

Violence can result in a high likelihood of injury, death, psychological harm, maldevelopment or deprivation [5]. Violence casts significant impact on adolescents during the critical time in their development when they experience rapid physical, cognitive and psychosocial growth and it may therefore affect the way they interact with the world around them [6]. In 2016, World Health Organisation (WHO) prepared INSPIRE: Seven strategies for ending violence against children, which is a technical package for actors committed to preventing and responding to violence against children and adolescents. It was developed in collaboration with the United States Centres for Disease Control and Prevention (CDC), the Global Partnership to End Violence Against Children, and other partners [7]. Furthermore, due to the decades-long neglect of adolescents in national health plans, the WHO and their partners produced Global Accelerated Action for the Health of Adolescents (AA-HA!): Guidance to Support Country Implementation in 2017; this guidance includes an analysis of interpersonal violence and suggests approaches which could be used to respond to the violence [8]. The guidance is expected to help countries implement the Global strategy for women's, children's and adolescents' health (2016–2030), and it is hoped it will support the achievement of the Sustainable Development Goals (SDGs) in each country. Thus, the global community has been proactive in shedding light on and improving adolescent health as well as supporting the efforts of countries to promote the protection of adolescents against violence during their formative years.

A report from 2019 shows that globally one third of adolescents aged 13–15 years were involved in a physical fight in the previous year [9]. A recent study also found that over one third of adolescents aged 12–15 years in low- and middle-income countries experienced having been physically attacked more than once during the past 12 months as well as some form of physical fighting; the prevalence was higher among boys than girls for having been physically attacked (41%, 29%) and physical fighting (46%, 27%) respectively [3]. According to the previous studies, the Region of the Americas marked the lowest prevalence of physically attacked among adolescents and the second lowest prevalence of physical fighting after the South-East Asia Region [3,10]. The country studies, using the same method of analysis, presented that Chile and El Salvador had a lower prevalence (11.5% and 13.1% respectively) of physical fighting among school-attending adolescents than Namibia, Pakistan, and Kuwait

(16.9%, 20.0%, 25.2% respectively) [11–15]. A segment of prior studies of several countries including Brazil, Ghana, Kuwait, and Spain also found that parental support and helpful peers were regarded as protective attributes [13,16–18].

Paraguay is located at the heart of South America and is surrounded by Argentina, Bolivia, and Brazil. Although the country is categorised as an upper-middle-income country, approximately a quarter of the total population remains below the official poverty line [19]. In 2012, the total secondary school enrolment rates were 65.9%, 65.8% for males, and 66.0% for females [20]. The Global Burden of Disease (GBD) study shows that interpersonal violence has been one of the leading causes of deaths and disability-adjusted life years (DALYs) among Paraguayan adolescents, and at much higher rates than the global average [4]. However, a previous report demonstrated that 20.2% of adolescents aged 13–15 years in Paraguay had experienced at least one physical fight in the past 12 months [4,9]. Among the available and comparable data, this percentage was ranked as the lowest in the Region of the Americas and the fifth lowest prevalence globally, following Lao People's Democratic Republic, Cambodia, China, and Swaziland (Eswatini) [9]. This study aims to investigate the most recent prevalence of physical fighting among school-attending adolescents in Paraguay in order to identify problematic fighting behaviour. We also aim to examine the correlates of physical fighting and the extent to which previously identified individual-, family- and peer-level factors are associated with physical fighting among school-attending adolescents in the country.

## Materials and methods

### Data sources

Global School-based Student Health Surveys (GSHS) are nationally representative surveys. The World Health Organization (WHO) and the Centers for Disease Control and Prevention (CDC), in the United States have developed the methodology for the GSHS. This methodology involves cross-sectional surveys using a self-administered questionnaire. GSHSs employ a two-stage cluster sample design to produce nationally representative data for all students in the country. The first stage entails the selection of the schools with a probability proportional to the enrolment size, and, in the second stage, classes are randomly selected. All students present in the selected classes on the day of the survey administration are eligible to participate in the survey. Data are publicly available on the WHO and CDC websites. Data from the nationally representative survey conducted in 2017 in the Republic of Paraguay were used for secondary analysis in this study. Detailed information on the data collection methods, the questionnaire, and the procedure are available on the website (http://www.cdc.gov/gshs/). The survey was administered to adolescents attending the schools and collected self-reported information on various health and behavioural indices. In Paraguay, 3,149 students at the age of 13–17 years completed the survey questionnaire; the school, student, and total response rates were 100%, 87%, and 87%, respectively.

The analyses presented in this paper did not exclude any cases in order to ensure a correct design-based analysis. Missing values were not imputed, and when reporting the results for a single variable, all available data were analysed. For bivariate and multivariate analysis, complete-case analysis is reported. Information on age was missing in 48 of the records, 62 records did not have information on sex, and both age and sex were missing 21 records; in addition, 18 records had no information on physical fights.

### Measurements

Following the previous studies which studied the physical fighting among school-attending adolescents using GSHS data from various other countries as well as data availability, we

included physical fighting as a dependent variable and sixteen other independent variables [3,11–16,21–25]. Physical fighting as a dependent variable was derived from one question in the GSHS:

> "*During the past 12 months, how many times were you in a physical fight*?"

The response options range was from "0 times", "1 time", "2 or 3 times", "4 or 5 times", "6 or 7 times", "8 or 9 times", "10 or 11 times" or "12 or more times." For the purpose of our analyses, participants were classified as having participated in a physical fight if they reported being involved in two or more fights (N = 251). We classified fighting behaviour into what could be considered potentially problematic (which has the possibility to result in short or long term sequalae) and potentially non-problematic (rough play). The aim with the classification lies in avoiding pathologizing adolescent development patterns, particularly the behaviour of adolescent males who may display higher levels of physical aggression, but this aggression does not in and of itself constitute problematic behaviour [26,27]. If no involvement in fight or incident was reported, participants were classified as not participating in a physical fight (N = 2,880).

We investigated twelve independent variables at the individual level (age, sex, anxiety, suicide planning, loneliness, truancy, bullying victimization, physical activity, sedentary, early sexual debut, alcohol use, and having been physically attacked) and four independent variables at the social level (presence of supportive parental figures, presence of helpful peers, extent of social network, and food insecurity). There were only twenty-three students aged 11 years or younger and, owing to this small number, they were combined with the next age group i.e. the 12 years age group. Details on how these variables were created are provided in Table 1.

## Statistical analysis

Due to the dichotomous nature of the outcome variable 'physical fighting (no = 0, yes = 1), multivariable binary logistic regression models were developed to identify the factors associated with physical fighting [28]. First, the distribution of selected independent variables in the entire sample was examined, followed by the distribution of selected independent variables within the dichotomised physical fights involvement variable. Odds ratios and their statistical significance, using the significance level of less than five percent ($p < 0.05$), for the association between involvement in a physical fight and the independent variables were analysed using a survey version of binary simple logistic regression models.

This step was followed by two additional survey binary multivariable logistic regression models. These were intended to model the ability of the selected independent variables to determine the association with the dichotomised variable for involvement in physical fights. The first set of models adjusted only for age and sex, while the second model included all those variables that were found to be statistically significant at the bivariate level. The choice of variables in the final logistic regression model were based on the results of bivariate analyses. Multicollinearity between all the explanatory variables and the Goodness-of-Fit test for the final multivariable model was verified. The measures of association are reported as adjusted and unadjusted odds ratios and associated 95% confidence intervals (CI).

Design-based analyses were carried out using Stata 16 program (StataCorp, 2019), by taking into account the complex survey design. All proportions, results of chi-square tests, and logistic regression models are reported based on design-based analysis; while unweighted counts/ frequencies are reported.

**Table 1. Independent variable derivation from Paraguay GSHS survey data 2017.**

| Survey Question | Coding | Variable |
|---|---|---|
| | **Individual-level variables** | |
| How old are you? | 13–18 years, aged less than 11 and 12 coded as aged 13. 18 years and older coded as 18. (coded continuous) | Age |
| What is your sex? | Male (1) Female (0) | Sex |
| During the past 12 months, how often have you been so worried about something that you could not sleep at night? | Most of the time/always (1) Never/rarely/sometimes (0) | Anxiety |
| During the past 12 months, did you make a plan about how you would attempt suicide? | Yes (1) No (0) | Suicide Plan |
| During the past 12 months, how often have you felt lonely? | Most of the time/always (1) Never/rarely/sometimes (0) | Loneliness |
| During the past 30 days, how many days did you miss classes or school without permission? | 0–2 times (0) 3 or more days (1) | Truancy |
| During the past 30 days, on how many days were you bullied? | 0 times (0) 1 or more times (1) | Bullying |
| During the past 7 days, on how many days were you physically active for a total of at least 60 min per day? | 3 days or less (0) 4 days or more (1) | Physical activity |
| How much time do you spend during a *typical* or usual day sitting and watching television, playing computer games, talking with friends, or doing other sitting activities? | 2 h or less (0) 3 h or more (1) | Sedentary |
| How old were you when you had sexual intercourse for the first time? | Never had sex or had after age 14 (0) Had sex at age 14 or earlier (1) | Early sexual debut |
| During the past 30 days, on how many days did you have at least one drink containing alcohol? | 0 days (0) 1 or more days (1) | Alcohol use |
| During the past 12 months, how many times were you physically attacked? | 0 time (0) 1–12 or more times (1) | Physically attacked |
| | **Social-level variables** | |
| During the past 30 days, how often did your parents or guardians understand your problems and worries? | Most of the time/always (1) Never/rarely/sometimes (0) | Supportive parental figures |
| During the past 30 days, how often were most of the students in your school kind and helpful? | Most of the time/always (1) Never/rarely/sometimes (0) | Helpful peers |
| How many close friends do you have? | 0 close friends (0) 1 close friends (1) 2 close friends (2) 3+ close friends (3) (coded continuous) | Close friends |
| During the past 30 days, how often did you go hungry because there was not enough food in your home? | Most of the time/always (1) Never/rarely/sometimes (0) | Food insecurity |

## Results

Within the recall period, 8.0% (unweighted count: 251) of the participants reported being involved in two or more physical fights; among the males and females the percentage of physical fighting involvement was 11.4%, and 4.7%, respectively. Table 2 shows the weighted cumulative distribution, as well as the distribution of physical fight involvement status in terms of percentages for all of the sixteen factors studied. For the adolescents' age and the number of close friends, mean and standard deviations are provided. Approximately half (48.8%) of the respondents were males. Overall, the two most common prevalent attributes were having helpful peers (62.8%) and having supportive parental figures (49.2%). While the two least common prevalent attributes were food insecurity (2.5%), and truancy (4.1%).

**Table 2. Cumulative proportion of factors in school-attending adolescents in Paraguay, GSHS 2017.**

| Variable | Cumulative Percentage [Unweighted Count] | Percent Not involved in physical fights [*Unweighted Count] | Percent Involved in physical fights [*Unweighted Count] | p-value |
|---|---|---|---|---|
| Age (SD) | 14.90 (1.58) [3,101] | 14.90 (1.59) [2,838] | 14.85 (1.43) [247] | 0.685 |
| Sex (Male) | 48.81 [3,087] | 46.94 [2,825] | 69.75 [247] | <0.001 |
| Anxiety | 9.47 [3,131] | 9.00 [2,866] | 15.15 [249] | 0.006 |
| Loneliness | 11.13 [3,090] | 10.52 [2,837] | 17.62 [241] | 0.004 |
| Food deprivation | 2.45[3,113] | 2.45 [2,854] | 2.53 [245] | 0.929 |
| Close friends (SD) | 2.48 (0.89) [3,101] | 2.47 (0.89) [2,844] | 2.54 (0.89) [243] | 0.263 |
| Bullying victimization | 16.76 [3,974] | 15.41 [2,731] | 32.59 [231] | <0.001 |
| Truancy | 4.07 [3,109] | 3.59 [2,847] | 9.23 [248] | <0.001 |
| Physical Activity | 33.71 [3,066] | 32.12 [2,805] | 52.88 [247] | <0.001 |
| Sedentary | 34.02 [3,089] | 33.89 [2,837] | 35.85 [241] | 0.526 |
| Supportive parental figures | 49.15 [3,099] | 50.16 [2,842] | 37.58 [243] | 0.002 |
| Helpful peers | 62.81 [3,077] | 63.38 [2,824] | 56.03 [240] | 0.026 |
| Suicide planning | 13.28 [3,085] | 12.14 [2,830] | 25.42 [242] | <0.001 |
| Early sexual debut | 11.43 [3,41] | 10.20 [2,792] | 26.14 [233] | <0.001 |
| Alcohol use | 35.27 [3,40] | 33.11 [2,781] | 60.07 [244] | <0.001 |
| Attacked | 15.41 [3,101] | 12.82 [2,841] | 44.37 [248] | <0.001 |

Notes

All variables are expressed as percentages with the exception of age and close friends, which are expressed as mean and standard deviation.

The square brackets show the unweighted counts, while all the reported percentages are weighted; the results of tests and their p-values are all based on the survey versions of Pearson chi-square tests, which accounts for the complex survey design, to examine the differences in % involvement in physical fight status across each variable.

The total number of respondents was 3,149. Differences across each variable are due to missing records in the variables.

*For involvement in physical fights 18 records were missing, as such the totals for each variable by physical fighting status are a little different from the totals reported in these two columns. This is because these two columns show the results where the variable, as well as the physical fighting status records, are available.

Column 2 of Table 3 shows the odds ratios for the individual association of involvement in physical fights with all sixteen selected factors, and their statistical significance using simple binary logistic regression models. The results of the bivariate analyses show that of the sixteen attributes, four i.e. age, food insecurity, sedentary lifestyle, and number of close friends were not statistically significantly associated with having been involved in physical fights in the past 12 months.

Column 5 of Table 3 shows the analysis of all the attributes studied adjusted for age and sex. The sex and age variables were each, adjusted for the other. Statistically significant associations were found between twelve of the sixteen attributes; the four attributes that were not statistically significantly associated with physical fighting were age, food insecurity, number of friends, and sedentary lifestyle.

Column 8 of Table 3 provides results of the final multivariable model i.e. after adjusting for all the covariates that were found to be statistically significant in the bivariate analysis, as reported in column 4 of Table 3. As a result, a total of 2,520 observation were used in the final multivariable logistic regression model, for information was available for all variables. Of the twelve attributes, six i.e. sex, physical activity, suicide planning, early sexual debut, alcohol use, and having been physically attacked were found to be statistically significantly associated with having been involved in physical fights at the p-value of <0.05 as well as <0.01. The goodness-of-fit test revealed that this was a good multivariate logistic model for the involvement in physical fighting by Paraguayan school-attending students ([F: 9, 17] 1.75; p-value: 0.1532).

**Table 3. Unadjusted odds ratios, adjusted odds ratios for age and sex, and adjusted odds ratios for all statistically significant variables for the association of physical fights with selected attributes among school-attending adolescents in Paraguay, GSHS 2017.**

| Variables | Simple Logistic Regression Analysis | | | Multivariable Logistic Regression Analysis—Adjusted for Age and Sex (Model 1*) | | | Multivariable Logistic Regression Analysis—Adjusted for all variables significant in simple logistic regression model (Model 2**) | | |
|---|---|---|---|---|---|---|---|---|---|
| | Unadjusted OR | 95% CI | p-value | Adjusted OR | 95% CI | p-value | Adjusted OR | 95% CI | p-value |
| Age | 0.98 | 0.87–1.10 | 0.688 | 0.97 | 0.86–1.09 | 0.542 | N/A | | |
| Sex (Male) | 2.61 | 2.05–3.32 | <0.001 | 2.66 | 2.10–3.38 | <0.001 | 2.45 | 1.87–3.20 | <0.001 |
| Anxiety | 1.81 | 1.20–2.72 | 0.006 | 2.15 | 1.44–3.21 | 0.001 | 1.04 | 0.49–2.22 | 0.911 |
| Loneliness | 1.82 | 1.23–2.70 | 0.004 | 2.20 | 1.48–3.27 | <0.001 | 1.31 | 0.79–218 | 0.288 |
| Food deprivation | 1.03 | 0.49–2.20 | 0.929 | 0.99 | 0.47–2.11 | 0.994 | N/A | | |
| Close friends | 1.10 | 0.92–1.31 | 0.299 | 1.04 | 0.86–1.25 | 0.695 | N/A | | |
| Bullying victimization | 2.65 | 2.03–3.47 | <0.001 | 2.55 | 1.92–3.80 | <0.001 | 1.31 | 0.97–1.76 | 0.077 |
| Truancy | 2.73 | 1.90–3.94 | <0.001 | 2.64 | 1.70–4.11 | <0.001 | 1.90 | 0.99–3.61 | 0.051 |
| Physical Activity | 2.37 | 1.77–3.18 | <0001 | 2.04 | 1.51–2.75 | <0.001 | 2.16 | 1.45–3.22 | <0.001 |
| Sedentary | 1.09 | 0.83–1.44 | 0.527 | 1.19 | 0.90–1.58 | 0.219 | N/A | | |
| Supportive parental figures | 0.60 | 0.44–0.81 | 0.002 | 0.55 | 0.40–0.75 | 0.001 | 0.90 | 0.62–1.31 | 0.568 |
| Helpful peers | 0.74 | 0.56–0.96 | 0.026 | 0.71 | 0.56–0.91 | 0.008 | 0.93 | 0.69–1.25 | 0.607 |
| Suicide planning | 2.47 | 1.90–3.20 | <0.001 | 2.98 | 2.34–3.80 | <0.001 | 1.89 | 1.28–2.80 | 0.003 |
| Early sexual debut | 3.11 | 2.32–4.19 | <0.001 | 2.64 | 1.94–3.59 | <0.001 | 1.90 | 1.28–2.84 | 0.003 |
| Alcohol use | 3.04 | 2.40–3.84 | <0.001 | 3.32 | 2.68–4.10 | <0.001 | 1.95 | 1.48–2.57 | <0.001 |
| Attacked | 5.42 | 4.07–7.23 | <0.001 | 4.97 | 3.67–6.72 | <0.001 | 3.29 | 2.17–4.99 | <0.001 |

OR, Odds Ratio; 95% CI, 95% Confidence Interval.

Note: Table 3 shows the results of design-based logistic regression models which took into account the complex survey design.

* All estimates are adjusted for age and sex; age and sex are each adjusted for the other.

** All estimates are adjusted for all variables listed in the model 2.

## Discussion

This study highlights the prevalence and correlates of physical fighting among school-attending adolescents in Paraguay. Our analyses indicated that 8.0% of the survey participants (11.4% in males, and 4.7% in females) were involved in two or more physical fights during the past 12 months in 2017. In the multivariable model, having been physically attacked, male gender, physical activity, alcohol use, early sexual debut, and suicide planning were statistically significantly associated with involvement in physical fighting. In the models adjusted for age and sex, age, or sex, the two most protective associations were the attributes of having helpful peers and supportive parents, however, these were not found to be statistically significant in the final model adjusted for all variables.

The Region of the Americas is likely to show a lower prevalence of problematic physical fighting among adolescents than other regions and Paraguay also shows a lower prevalence than countries in the other regions [13–15]. This regional trend is also true even when compared to the study results about more frequent physical fights such as over four or 12 physical fights per year [10,25]. Within the region, Paraguayan adolescents experienced more frequent physical fighting than those in El Salvador and less fighting than those in Chile [11,12]. Even compared to a study with the higher threshold of physical fights, the prevalence in Paraguay is likely to be in the lower groups within the same region: for example, for males, there was an 11.4% prevalence in two or more physical fights in Paraguay; in four or more physical fights it was 4.5% in Costa Rica, 7.5% in Colombia, 8.7% in Venezuela, and 11.4% in Peru in the upper

middle-income group; 12.4% in Uruguay, 13.8% in Chile, and 17.7% in Jamaica in the high-income group [10].

Consistent with other studies, boys are more likely to report frequent fighting than girls [10,21,25,29]. However, the magnitude of the difference by gender varies among countries. Paraguayan boys are 2.5 times more likely to be involved in physical fighting compared to girls, whereas in Chile this becomes 2.9 times and 3.6 times in El Salvador, both of which are in the same region as Paraguay. It is 2.1 times in Namibia, which is in the upper-middle-income group like Paraguay [11,12,15]. This study also found that being more physically active was associated with engaging in more physical fighting involvement. This aligns with the studies conducted in El Salvador and Kuwait where physical inactivity might lead to less physical fighting but is in contrast to the findings of a study of European and North America countries where physically active adolescents were more likely not to be involved in fights [12,13,30]. Mahalik and colleagues suggested that traditional masculine gender socialisation and social norms encourage boys to put their health at risk such as exposing themselves to more physical fights [31]. Looze and colleagues found that boys reported higher levels of fighting and physical activity in more gender-unequal countries than boys in more gender-equal countries [22]. Paraguay was found to maintain clear gender gaps at personal and political levels and the latest gender inequity index for Paraguay was 0.5 [32,33]. However, other studies reported that physical activity was associated with less involvement in other health risk behaviours such as alcohol and drug use, and that less involvement in high-risk behaviours had links with less involvement in physical fighting [34,35]. This suggests that more physical activity might bring less physical fighting. The content and role of physical activity as well as the contexts around the physical activity of adolescents in each country need to be further investigated.

Our study indicated that alcohol use, early sexual debut, and suicide planning had similar levels of association with physical fighting in Paraguay. Adolescents who had any of these attributes were almost twice as likely to experience physical fighting. Consistently, many studies from other countries including the United States, Malaysia, Ghana, and Sri Lanka revealed that adolescents who drink alcohol were more at risk of engaging in physical fights [16,23,24,36–38]. Early sexual debut was also found to be associated with physical fights according to multi-country studies [39,40]. The studies in Brunei, El Salvador, Namibia, Pakistan, Kuwait, and the USA have reported that adolescents experiencing suicide planning and other psychosocial problems were more likely to be involved in frequent physical fighting [12–15,41–43]. Moreover, other studies identified the associations between alcohol use, early sexual debut, and suicide planning [44,45].

The global community recognises that, in order to make progress in complex areas such as prevention of violence, suicide, early sexual debut and/or pregnancy, and substance and/or alcohol use in adolescent health, intersectoral programmes and collaboration are effective [8]. According to the progress report of INSPIRE which aims to end violence against children and adolescents, Paraguay has shown better progress when compared to other upper middle-income countries in the same regions in the following areas; implementation and enforcement of laws, parent and caregiver support, and education and life skills such as school enrolment and school-based prevention [46]. However, Paraguay lags behind such areas as the strengthening of social norms and values, responses such as mental health services for both victims and perpetrators, identification and referral of perpetrators to the legal system, and reducing violence by school staff [46]. These areas need to be strengthened along with continuous improvement of other areas.

Adolescents having helpful peers and supportive parental figures were found to be at less risk from physical fighting involvement in Paraguay in the unadjusted and age-and-sex-adjusted analyses in this study. However, after adjusting all the variables, these two variables

were not statistically significant. This is not in line with prior studies of several other countries including Brazil, Spain, the USA, Kuwait, and Ghana, where parental support and helpful peers were deemed protective attributes [13,16–18,47]. However, many families in Paraguay tend to regard parental corporal punishment as a teaching measure. Although the Law to prohibit all corporal punishment of children was approved and enacted in Paraguay in 2016 as the 50th state worldwide and the 10th state in the region, the tradition has remained [48]. Elgar and colleagues indicated that, compared to 20 countries with no ban, 30 countries with bans on corporal punishment in schools and at home experienced 31% and 58% less physical fighting in adolescent males and females respectively [49].

Parental support is seen as a preventive attribute in the studies as mentioned above. Most of the countries in the Region of the Americas have enacted legislation banning corporal punishment and several countries—other than Paraguay—have shown good progress in strengthening social norms and values [46]. Given these findings, Paraguay will also be able to make changes in the relationship between parents and children by better targeted and sustained awareness-raising campaigns and better enforcement of the legislation.

Regarding helpful peers, the scarcity of the data should be noted. In Paraguay in 2011, the rate of survival to the last grade of primary school was 84.2% and the rate of effective transition rate from primary to lower secondary general education was 94.5% [50]. However, since 2013 the data available on the completion of secondary school as well as the data on the progress and completion of education has been scarce. Therefore, an examination of the peer-related environment needs to find more sources in order to develop the discussions.

This study found that among all the explanatory variables, having been physically attacked shows the strongest association with physical fighting, while the association between being bullied and physical fighting was not statistically significant in the analysis adjusted for all other explanatory variables. Owing to the cross-sectional nature of the survey, it is not possible to determine if adolescents were involved in physical fighting as a defence against being attacked, however, a multicountry study suggests that countries in the Region of the Americas tend to have similar prevalence in having been physically attacked and physical fighting involvement, which countries from other regions do not necessarily have [3]. This might support the association of these two attributes in this study.

Lastly, considering interpersonal violence as a leading cause of deaths and DALYs, other kinds of interpersonal violence including bullying, emotional abuse, sexual and physical assault, and homicide, as well as their contexts, need to be assessed so that comprehensive strategies and preventive measures can be introduced to protect Paraguayan adolescents from problematic violence.

This study has several limitations. First, this is a cross-sectional study which captured only the responses of the students who attended school on the day of the survey. School attendance could be correlated with the risk of physical fighting and being physically attacked due to sequels of previous episodes or preventive behaviour. Second, the cross-sectional nature of the data does not allow causal interpretation of the associations studied. Reverse causality could also affect behavioural variables such as being physically attacked. Despite this, the study was able to suggest potential groups who were at a higher risk of being involved in physical fighting. Third, this study used a self-report survey. This might have introduced a social-desirability bias and a relevant bias into the survey results. It includes fears of possible implications of declaring attacks. Fourth, dichotomised variables could result in less precise measurements. Lastly, the reasons, triggers or consequences of physical fighting cannot be identified in the study. However, this is the first study to examine the prevalence and correlates of physical fighting among school-attending adolescents in Paraguay. We also used the most recent survey data of 2017 from a nationally presentative survey. The findings of this study could be useful

for policy makers to develop health strategies and plans for adolescent health and in accelerating preventive measures for problematic physical fighting.

## Conclusions

This study describes problematic physical fighting behaviour among adolescents in Paraguay and the associated factors. Although Paraguay shows relatively lower prevalence of physical fighting than other countries, the high association between physical fighting involvement and having been physically attacked is noteworthy. Considering that serious interpersonal violence is causing deaths among Paraguayan adolescents, other kinds of violence including bullying, emotional abuse, sexual and physical assault, and homicide, as well as their backgrounds, need to be assessed so that comprehensive strategies and measures can be introduced to protect Paraguayan adolescents from problematic violence. Furthermore, in addition to the good progress in the implementation of laws to advance the prevention of violence, it is critical to strengthen social norms and values against violence as well as practical responses such as providing mental health services for both victims and perpetrators. To support the prevention of violence, the bans on corporal punishment in schools and at home and the changes in the relationship between parents and children need to be strongly promoted by better awareness-raising and enforcement of the legislation.

## Supporting information

**S1 Table. Table 1: Independent variable derivation from Paraguay GSHS survey data 2017.**
(DOCX)

**S2 Table. Table 2: Cumulative proportion of factors in school-attending adolescents in Paraguay, GSHS 2017.**
(DOCX)

**S3 Table. Table 3: Unadjusted odds ratios, adjusted odds ratios for age and sex, and adjusted odds ratios for all statistically significant variables for the association of physical fights with selected attributes among school-attending adolescents in Paraguay, GSHS 2017.**
(DOCX)

## Author Contributions

**Conceptualization:** Hiroko Taniguchi, Masood Ali Shaikh.

**Data curation:** Hiroko Taniguchi, Masood Ali Shaikh.

**Formal analysis:** Hiroko Taniguchi, Masood Ali Shaikh.

**Methodology:** Hiroko Taniguchi, Masood Ali Shaikh.

**Project administration:** Hiroko Taniguchi, Masood Ali Shaikh.

**Resources:** Hiroko Taniguchi, Masood Ali Shaikh.

**Validation:** Hiroko Taniguchi, Masood Ali Shaikh.

**Writing – original draft:** Hiroko Taniguchi, Masood Ali Shaikh.

**Writing – review & editing:** Hiroko Taniguchi, Masood Ali Shaikh.

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
