## [Decision Letter · Decision Letter 0]

22 Dec 2021

PONE-D-21-03008Physical fighting and associated factors among adolescents in Paraguay the 2017 national school-based health surveyPLOS ONE

Dear Dr. Hiroko Taniguchi

Thank you for submitting your manuscript to PLOS ONE. After careful consideration, we feel that it has merit but does not fully meet PLOS ONE’s publication criteria as it currently stands. Therefore, we invite you to submit a revised version of the manuscript that addresses the points raised during the review process.

ACADEMIC EDITOR: Please insert comments here and delete this placeholder text when finished. Be sure to:While the article is of interest, I agree with the reviewers that before it can be accepted for publication, it would require substantial revision particularly in terms of English and grammar corrections as well as alternate statistical analysis such as conducting Pearson’s chi-square test to check if the characteristics of those involved in physical fights were different from those who were not."  Please attend to the reviewers comments

We look forward to receiving your revised manuscript.

Kind regards,

Valsamma Eapen, MBBS, PhD, FRCPsych, FRANZCP

Academic Editor

PLOS ONE

Journal Requirements:

2. Please refer to the specific statistical analyses performed as well as any post-hoc corrections to correct for multiple comparisons. If these were not performed please justify the reasons. Please refer to our statistical reporting guidelines for assistance (https://journals.plos.org/plosone/s/submission-guidelines.#loc-statistical-reporting)

Additional Editor Comments (if provided):

Dear Author,

While the article is of interest, I agree with the reviewers that before it can be accepted for publication, it would require substantial revision particularly in terms of English and grammar corrections as well as alternate statistical analysis such as conducting Pearson’s chi-square test to check if the characteristics of those involved in physical fights were different from those who were not."

Reviewers' comments:

Reviewer's Responses to Questions

**Comments to the Author**

1. Is the manuscript technically sound, and do the data support the conclusions?

Reviewer #1: Yes

Reviewer #2: Partly

2. Has the statistical analysis been performed appropriately and rigorously? 

Reviewer #1: Yes

Reviewer #2: No

3. Have the authors made all data underlying the findings in their manuscript fully available?

Reviewer #1: Yes

Reviewer #2: Yes

4. Is the manuscript presented in an intelligible fashion and written in standard English?

Reviewer #1: Yes

Reviewer #2: No

5. Review Comments to the Author

Reviewer #1: Please see the attached pdf file for Table format

Manuscript: Physical fighting and associated factors among adolescents in Paraguay the 2017 national school-based health survey

Specific comments are:

#1 Page-4, Ln 75: ‘Accelerated Action for the Health of Adolescents (AA-HA!) –there is a typo in bracket – delete the sign (!).

# 2 Page-5, Ln 84-87: “A recent study… over one third of adolescents …, it comes to boys, the prevalence increased to 41% in physically attacked and 46% in physical fighting”. It looks rates for boys are substantially higher than overall rates of one third – in this instance I recommend to specify rates for both girls and boys respectively– that will be helpful for readers and policy makers to take initiatives to reducing the gender gap.

#3 Page-5, Ln 100: “… causes of deaths and DALYs …”. DALYs need to be spelled out at least once.

#4 Page-6, Ln 104: “... following Lao PDR, Cambodia …”. PDR need to be spelled out.

#5 Page-8, Table 1: “How many close friends do you have? 0 close friends (0) 1 close friends (1) 2 close friends (2) 3+ close friends (3) (coded continuous)” – since there are only 3 values – author can say it is categorical.

#6 Page-9, Ln 155-156: “Similar to previous studies, multivariable logistic regression models were developed to identify factors associated with physical fighting [10-14]. Instead of referring readers to look at five previous studies [10-14] – the author can simply say due to dichotomous nature of the outcome variable ‘physical fighting (no=0, yes=1), multivariable binary logistic regression models were developed to identify factors associated with physical fighting [give reference of Stat Book; e.g. Applied Logistic Regression by David W. Hosmer, Rodney X. Sturdivant and Stanley Lemeshow]

#7 Page-9, Ln 159-160: “Odds ratios and their statistical significance, using the significance level of less than five Percent…”. The author can include (p<0.05) i.e. ‘five Percent’ should be ‘five percent (p<0.05)’.

#8 Page 9-11, Results: The authors used unweighted sample for basic prevalence: 8.0% (unweighted count: 251)….; among males and females the proportion of physical fighting involvement was 11.4%, and 4.7%, respectively. Whereas overall sample characteristics in Table 2 is based on weighted data: Column 2 and overall Table 2 is confusing – does not match with actual prevalence for subgroups. In Statistical analysis section - authors did not mention anything about data weighting – how the weights were calculated. As the methods section did not mention about weighting – it is assumed that the Multivariable logistic regression analysis was done based on unweighted data; and so did the unadjusted Odds Ratios (ORs). If the ORs are based on unweighted data – I strongly recommend to presented unweighted data in Table 2. Otherwise authors need to mention in methods section how data weighting constructed for both descriptive and logistic regression analysis; in that case prevalence should be based on weighted data.

#9 Page 12, Ln 193-94: “.. using simple binary logistic regression models”. This should be “… simple univariate binary logistic regression models”.

#10 Page 12, Ln 194-95: “The results of bivariate analyses show that out of sixteen attributes, four were not statistically significantly associated …”. The author did not show the bivariate analysis for prevalence of physical fights with significant labels. I recommend authors to reformat the Table 2 in following format based on unweighted data - including both sample characteristics and prevalence of physical fights for each variable category with p-values (bivariate analysis).

Suggested format for Table 2: Sample characteristics and percentage involved in any physical fights with selected attributes among school-attending adolescents in Paraguay, GSHS 2017 [see the attached pdf file]

#11 Page 13, Title “Table 3: Unadjusted odds ratios for the association …Paraguay, GSHS 2017”.

The author can reward the title as “Table 3: … odds ratios from univariate binary logistic regression models for the association … Paraguay, GSHS 2017”.

#12 Page 13, Table 3: Reference category for none of the variables are indicated. Reference category for each variable should be included.

#13 Page 13, Table 3: In variable column - SD included with Age [Age(SD)] and Close friends [Close friends (SD)]; male added with Sex [Sex (male)] – this typos need to be corrected.

#14 Page 13, Table 3: In footnote indicated “All estimates are adjusted for age and sex; age; or sex”.

This is confusing because these ORs are unadjusted from univariate binary logistic regression analysis.

#15 Page 15, Title “Table 4: Multivariate analysis…. adolescents in Paraguay, GSHS 2017”.

There are many types of multivariate analysis. It’s need be specific that “Multivariable logistic regression analysis…. adolescents in Paraguay, GSHS 2017”

#16 Page 15, Table 4: Reference category for none of the variables are indicated. Reference category for each variable should be included.

#17 Page 15, Table 4: In variable column - SD included with Age [Age(SD)] and Close friends [Close friends (SD)]; male added with Sex [Sex (male)] – this typos need to be corrected.

#18 Page 16, Table 4: In footnote indicated “All estimates are adjusted for age and sex; age; or sex”.

This is confusing because this kind of adjusted analysis for “age and sex”; “age; or sex” -requires series of multivariable logistic regression models. I believe authors did just one Multivariable logistic regression analysis. So the footnote should be “All estimates are adjusted for all variables included in the Multivariable logistic regression model”.

#19 Page 17, Title “Table 5: Outcomes of multivariable analysis of variables …., GSHS 2017”.

There are many types of multivariable analysis. So the authors need be specific that “Multivariable logistic regression analysis…., GSHS 2017”.

#20 Page 17, Table 5: Reference category for none of the variables are indicated. Reference category for each variable should be included.

#21 Page 13, 15, 17: Tables 3 to 5: Instead of using three Tables on ORs – the authors can presents results of Tables 3, 4 and 5 as one Table – so the readers can see the differences of unadjusted and adjusted ORs side by side in same Table with following format below. The new Table can be titled as

“Table 3: Unadjusted odds ratios (ORs) from univariate logistic regression analysis and adjusted ORs from multivariable logistic regression analysis with 95% confidence interval (95%CI) for the association … Paraguay, GSHS 2017 [[see the attached pdf file]

Reviewer #2: Title of the manuscript: Physical fighting and associated factors among adolescents in Paraguay the 2017 national school-based health survey

Thank you for the opportunity to review this paper. The aim of this study was to investigate the prevalence and correlates of physical fighting among adolescents in Paraguay based on the 2017 national school-based health survey. The manuscript addresses an interesting topic; however, there are some issues that need to be improved prior to publication. I believe this manuscript can be considered for publication after addressing the comments below:

Title: Physical fighting and associated factors among adolescents in Paraguay the 2017 national school-based health survey.

I suggest modifying the title to “Prevalence and correlates of Physical fighting among adolescents in Paraguay: Findings from the 2017 national school-based health survey.”

Abstract:

1. The English grammar and style should be checked throughout the manuscript.

Background: Line 28 – Causes of deaths – spelling error. Please change to causes of death.

2. Lines 30-31 – Sentence seems unfinished. The extent to which previously identified factors ..what?

3. Lines 43-44 – Please combine as one sentence.

Introduction:

1. The English grammar and style should be checked throughout the manuscript.

2. Lines 83-84 “tells us that” – too colloquial? Perhaps change to something formal.

3. Lines 87 – “41% in physically attacked and 46% in physical fighting” – word missing - change to “41% in being physically attacked and 46% involved in physical fighting”

4. Line 102 – change to “at least one physical fight in the last 12 months”

Methods:

1. No mention of study design used. I strongly recommend the authors to refer to STROBE checklist for cross-sectional/observational studies https://www.strobe-statement.org/

2. Key information such as number of schools, age of the participants included in the analysis and the eligibility criteria are missing in this section.

Lines 122-123 – Combine both sentences.

3. Lines 135-138 – What is the rationale behind classifying two or more fights as having participated in a physical fight and not one or more fights?

The sentence “two or more fights as having participated in a physical fight” in itself is contradictory.

4. Line 163 – Incomplete sentence. Please check grammar and sentencing throughout.

5. Statistical analysis must talk about descriptive analysis of baseline characteristics (presented in Table 2) followed by the main analysis.

6. How was multicollinearity checked? Please elaborate with the type of analysis.

Results:

1. Table 2 – Indicate number of participants along with percentages for each category and the number with missing data for each variable of interest. Please refer to the STROBE checklist.

2. I suggest conducting Pearson’s chi-square test to check if the characteristics of those involved in physical fights were different from those who were not.

3. Lines 206 – 12 out of 16 variables were significantly associate – not 14.

4. Line 217 – Typo – Uppercase “T” for table 3.

5. Table 5 – Sex to be modified to Sex (Male).

6. Tables 2-5 – I would suggest showing both categories for each variable for clarity and readability. For example, in Table 5,

Currently, it is not a table with standalone information, we need refer back to Table 1.

Variable Adjusted OR 95% CI p-value

Anxiety

Never/rarely/sometimes 1.00 Reference category

Most of the time/always 1.04 0.49-2.22 0.911

Discussion:

1. Lines 252-253 – Rephrase the sentence to “Consistent with other studies,…”

2. Lines 274 – I would suggest using the word “association” or “odds” given a regression analysis was conducted rather than use of term “correlation”

3. Lines 332-335 – Please expound on what the other sorts of interpersonal violence are based on your literature review.

4. Lines 351- Please change to …accelerate actions for “them”

5. The findings have been critically appraised with other literature/research findings, however, the “so what” factor, that is, implications of study findings to individuals/community and policy is vague.

Conclusion:

1. Lines 354-355 – Useful how? Please rephrase to say how the findings may be useful to inform policy and planning. What intervention programs are in place as best practice in Paraguay or other countries/ what is recommended in other countries?

6. PLOS authors have the option to publish the peer review history of their article (what does this mean?). If published, this will include your full peer review and any attached files.

Reviewer #1: No

Reviewer #2: **Yes: **James Rufus John

---

## [Author Response · Author response to Decision Letter 0]

5 Mar 2022

Manuscript ID: PONE-D-21-03008

Response to Reviewers

To

Dr Valsamma Eapen

Academic Editor

PLOS ONE

Dear Dr. Valsamma Eapen:

Thank you very much for your consideration of our manuscript, Physical fighting and associated factors among adolescents in Paraguay the 2017 national school-based health survey. We have revised the manuscript thoroughly according to the comments by the editor and reviewers. We addressed all points clearly in the revised manuscript.

Our responses to the editor and reviewers are provided in a separate sheet. Where we have changed the texts and the corresponding sentences in the texts have been highlighted using track changes. 

- The yellow-highlight: the changed parts responding to the reviewers’ comments.

- The grey-highlight: the sentences or clauses which include the changes based on the English-language editor’s corrections and/or suggestions. We used the professional English-language service for the revised manuscript.

All line numbers referred to are in the margins of our revised manuscript with tracked changes. 

We hope that the revisions are satisfactory in addressing issues raised by the editor and reviewers and look forward to hearing your decision about this article.

Yours sincerely,

Authors

Journal Requirements:

1. Please ensure that your manuscript meets PLOS ONE's style requirements, including those for file naming. The PLOS ONE style templates can be found at https://journals.plos.org/plosone/s/file?id=wjVg/PLOSOne_formatting_sample_main_body.pdf and https://journals.plos.org/plosone/s/file?id=ba62/PLOSOne_formatting_sample_title_authors_affiliations.pdf.

We checked the PLOS ONE's style requirements and accordingly revised the manuscript.

2. Please refer to the specific statistical analyses performed as well as any post-hoc corrections to correct for multiple comparisons.

Responding to the reviewers’ comments, we mentioned the models and tests we used in the manuscript.

3. We suggest you thoroughly copyedit your manuscript for language usage, spelling, and grammar. 

We, Hiroko Taniguchi and Masood Ali Shaikh, asked a professional English-language editor, Elizabeth Nyman, to edit the revised manuscript. We carefully checked every correction and/or suggestion and made further revisions to the manuscript.

Additional Editor Comments (if provided):

While the article is of interest, I agree with the reviewers that before it can be accepted for publication, it would require substantial revision particularly in terms of English and grammar corrections as well as alternate statistical analysis such as conducting Pearson’s chi-square test to check if the characteristics of those involved in physical fights were different from those who were not."

We have revised the manuscript thoroughly according to the comments by you and the reviewers. We addressed all points in the reponse below and the revised manuscript. 

Reviewer #1: Please see the attached pdf file for Table format

Manuscript: Physical fighting and associated factors among adolescents in Paraguay the 2017 national school-based health survey

Specific comments are:

#1 Page-4, Ln 75: ‘Accelerated Action for the Health of Adolescents (AA-HA!) –there is a typo in bracket – delete the sign (!).

This is the official name of the guidance. Please find the WHO website and the cover of the publication for your information. https://www.who.int/publications/i/item/9789241512343

# 2 Page-5, Ln 84-87: “A recent study… over one third of adolescents …, it comes to boys, the prevalence increased to 41% in physically attacked and 46% in physical fighting”. It looks rates for boys are substantially higher than overall rates of one third – in this instance I recommend to specify rates for both girls and boys respectively– that will be helpful for readers and policy makers to take initiatives to reducing the gender gap.

Thank you for your advice. We specified the rates for boys and girls. (Page 5, line 189-190)

#3 Page-5, Ln 100: “… causes of deaths and DALYs …”. DALYs need to be spelled out at least once.

We spelled it out. (Page 6, line 293)

#4 Page-6, Ln 104: “... following Lao PDR, Cambodia …”. PDR need to be spelled out.

We spelled it out. (Page 6, line 298)

#5 Page-8, Table 1: “How many close friends do you have? 0 close friends (0) 1 close friends (1) 2 close friends (2) 3+ close friends (3) (coded continuous)” – since there are only 3 values – author can say it is categorical.

The reason for calling this variable “coded continuous” is that in the logistic regression model we used it as a continuous variable. Therefore, in table 3 we provided only one Odds Ratio (OR) for this covariate owing to the fact that it was used as a continuous variable as opposed to categorial; in which case one would use one category as a reference and provide ORs for all others. Also in table 2, we have provided mean and SD for this variable. 

#6 Page-9, Ln 155-156: “Similar to previous studies, multivariable logistic regression models were developed to identify factors associated with physical fighting [10-14]. Instead of referring readers to look at five previous studies [10-14] – the author can simply say due to dichotomous nature of the outcome variable ‘physical fighting (no=0, yes=1), multivariable binary logistic regression models were developed to identify factors associated with physical fighting [give reference of Stat Book; e.g. Applied Logistic Regression by David W. Hosmer, Rodney X. Sturdivant and Stanley Lemeshow]

Thanks. In the revised manuscript we have incorporated the comment as under: (Page 9, line 581-583)

Due to the dichotomous nature of the outcome variable ‘physical fighting (no=0, yes=1), multivariable binary logistic regression models were developed to identify the factors associated with physical fighting [REFERENCE: David W. Hosmer Jr., Stanley Lemeshow, Rodney X. Sturdivant. Applied Logistic Regression. 3rd ed., John Wiley & Sons, 2013]. 

#7 Page-9, Ln 159-160: “Odds ratios and their statistical significance, using the significance level of less than five Percent…”. The author can include (p<0.05) i.e. ‘five Percent’ should be ‘five percent (p<0.05)’.

Thanks. In the revised manuscript we have incorporated the comment as under: (Page 9, line 586)

‘Odds ratios and their statistical significance, using the significance level of less than five percent (p<0.05), for the association between physical fight involvement with the independent variables were analyzed using survey version of binary simple logistic regression models.’

#8 Page 9-11, Results: The authors used unweighted sample for basic prevalence: 8.0% (unweighted count: 251)….; among males and females the proportion of physical fighting involvement was 11.4%, and 4.7%, respectively. Whereas overall sample characteristics in Table 2 is based on weighted data: Column 2 and overall Table 2 is confusing – does not match with actual prevalence for subgroups. In Statistical analysis section - authors did not mention anything about data weighting – how the weights were calculated. As the methods section did not mention about weighting – it is assumed that the Multivariable logistic regression analysis was done based on unweighted data; and so did the unadjusted Odds Ratios (ORs). If the ORs are based on unweighted data – I strongly recommend to presented unweighted data in Table 2. Otherwise authors need to mention in methods section how data weighting constructed for both descriptive and logistic regression analysis; in that case prevalence should be based on weighted data.

Kindly note that on:

Page 9, line 587-588, we stated that “… were analysed using a survey version of binary simple logistic regression models.” 

Page 9, line 590-591, we stated that “This step was followed by two additional survey binary multivariable logistic regression models.” 

Page 10, line 633-634, we stated that “All proportions – expressed in percentages – are weighted”.

The lines from the original manuscript submitted specify that all analyses involved survey versions of the logistic regression models and that all percentages reported are weighted. So, unless otherwise specified, we used design-based analyses. Hence all analyses incorporated – took into account – survey design of the GSHS.

Hence, we respectfully, do not concur with some of the comments specified here, and our responses to each statement are as under please:

(a) “The authors used unweighted sample for basic prevalence: 8.0% (unweighted count: 251)….; among males and females the proportion of physical fighting involvement was 11.4%, and 4.7%, respectively.”

We used a weighted sample and provided weighted prevalence i.e. 8.0%, 11.4%, 4.7%. However, we also provided an unweighted count of 251; which is the number of students who were involved in two or more fights. 

(b) “Column 2 and overall Table 2 is confusing – does not match with actual prevalence for subgroups.”

In this table, Column 2 provides cumulative i.e. overall percentages; so overall in the survey 48.81% of respondents were males. Column 3 i.e. ‘percent not involved in physical fights’ refers to the subgroup that was not involved in physical fights and in this subgroup males comprised 46.94%, while the rest were females. Column 4 i.e. ‘Percent involved in physical fights’ refers to the subgroup that was involved in physical fights, and in this subgroup, males comprised 69.75%. 

(c) “In Statistical analysis section - authors did not mention anything about data weighting – how the weights were calculated.”

Since this was a secondary analysis of the GSHS data. We used the survey design variables i.e. weights, strata and PSU, that were provided with the dataset and were calculated by the agencies that conducted this survey. Their detailed methodology is available on the website from where we downloaded this dataset. We provided the weblink in addition to stating that details on the methodology of the survey are also provided there. (Page 7, line 417)

(d) As the methods section did not mention about weighting – it is assumed that the Multivariable logistic regression analysis was done based on unweighted data; and so did the unadjusted Odds Ratios (ORs). If the ORs are based on unweighted data – I strongly recommend to presented unweighted data in Table 2. Otherwise authors need to mention in methods section how data weighting constructed for both descriptive and logistic regression analysis; in that case prevalence should be based on weighted data.

As stated above, we mentioned in the methods section that analyses are weighted. The “unadjusted Odds Ratios” refers to the use of simple logistic regression models as part of bivariate analysis i.e. the simple logistic regression models only had one explanatory/independent variable. As such the resulting odds ratios were unadjusted for any other explanatory variables. The multivariable models had more than one explanatory variable, and as such, each explanatory variable was adjusted for all the other explanatory variables in the models.

#9 Page 12, Ln 193-94: “.. using simple binary logistic regression models”. This should be “… simple univariate binary logistic regression models”.

On Page 12, lines 687-689, We stated that “Column 2 of Table 3 shows the odds ratios of the individual association of involvement in physical fights with all sixteen selected factors, and their statistical significance using simple binary logistic regression models. The results of the bivariate analyses show that …” 

If we revise the statement as per comment the second sentence (highlighted in bold above) would follow by talking about ‘bivariate analysis.’ We request that “simple binary logistic regression’ be kept as such, as the term “simple” logistic regression model implies that only one explanatory variable is in the model. 

#10 Page 12, Ln 194-95: “The results of bivariate analyses show that out of sixteen attributes, four were not statistically significantly associated …”. The author did not show the bivariate analysis for prevalence of physical fights with significant labels. I recommend authors to reformat the Table 2 in following format based on unweighted data - including both sample characteristics and prevalence of physical fights for each variable category with p-values (bivariate analysis).

Suggested format for Table 2: Sample characteristics and percentage involved in any physical fights with selected attributes among school-attending adolescents in Paraguay, GSHS 2017 [see the attached pdf file]

Table 2 has been revised as per the comment in the revised manuscript. (Page 11)

#11 Page 13, Title “Table 3: Unadjusted odds ratios for the association …Paraguay, GSHS 2017”.

The author can reward the title as “Table 3: … odds ratios from univariate binary logistic regression models for the association … Paraguay, GSHS 2017”.

The “univariate” implies analysis pertaining to one variable only. However, a simple logistic regression model with two variables i.e. one dependent and the other independent is a bivariate analysis. We used the term ‘unadjusted’ to clearly state that the analysis reported in this pertains of each explanatory variable’s association with the outcome variable. As such all coefficients are “unadjusted”. (Page 13)

#12 Page 13, Table 3: Reference category for none of the variables are indicated. Reference category for each variable should be included.

All explanatory variables used are binary (1/0 coded) variables. It is respectfully pointed out that when reporting binary variables, it is the statistical norm that presence/yes/1 is compared against absence/no/0. 

#13 Page 13, Table 3: In variable column - SD included with Age [Age(SD)] and Close friends [Close friends (SD)]; male added with Sex [Sex (male)] – this typos need to be corrected.

Thank you for pointing this out. In Table 3 in the revised manuscript these types have been corrected and deleted. However, “sex (male)” implies that males are compared against females, so we suggest keeping it as it is, please. (Page 13)

#14 Page 13, Table 3: In footnote indicated “All estimates are adjusted for age and sex; age; or sex”. This is confusing because these ORs are unadjusted from univariate binary logistic regression analysis.

Thanks indeed for pointing this out. This error was the result of copying/pasting from the original table 4, which was a mistake. In the revised manuscript this footnote is kept only for Column 3 (the original table 4). (Page 13, revised Table 3, line 705)

#15 Page 15, Title “Table 4: Multivariate analysis…. adolescents in Paraguay, GSHS 2017”.

There are many types of multivariate analysis. It’s need be specific that “Multivariable logistic regression analysis…. adolescents in Paraguay, GSHS 2017”

Thanks for pointing this out. In the revised manuscript, the “Multivariate” has been replaced with ‘Multivariable’. (Page 13, revised Table 3, Column 3)

#16 Page 15, Table 4: Reference category for none of the variables are indicated. Reference category for each variable should be included.

All explanatory variables used are binary (1/0 coded) variables. It is respectfully pointed out that when reporting binary variables, it is the statistical norm that presence/yes/1 is compared against absence/no/0. (Page 13, revised Table 3, Column 3)

#17 Page 15, Table 4: In variable column - SD included with Age [Age(SD)] and Close friends [Close friends (SD)]; male added with Sex [Sex (male)] – this typos need to be corrected.

Thank you for pointing this out. In the revised manuscript these types have been corrected and deleted. However, “sex (male)” implies that males are compared against females, so we suggest keeping it as it is, please. (Page 13, revised Table 3)

#18 Page 16, Table 4: In footnote indicated “All estimates are adjusted for age and sex; age; or sex”.

This is confusing because this kind of adjusted analysis for “age and sex”; “age; or sex” -requires series of multivariable logistic regression models. I believe authors did just one Multivariable logistic regression analysis. So the footnote should be “All estimates are adjusted for all variables included in the Multivariable logistic regression model”.

We did in fact run a series of multivariable logistic regression models. This was pointed out as below. 

The first set of models adjusted only for age and sex. (Page 9, line 592-593)

We also rewrote the footnote of Column 3 of Table 3 (originally Table 4) and relevant main text as below in order to make this point clearer:

*All estimates are adjusted for age and sex; age and sex are each adjusted for the other. (Page 13. line 705)

Column 3 of Table 3 shows the analysis of all the attributes studied for age and sex . The sex and age variables were each, adjusted for the other. (Page 14, line 708-709)

#19 Page 17, Title “Table 5: Outcomes of multivariable analysis of variables …., GSHS 2017”.

There are many types of multivariable analysis. So the authors need be specific that “Multivariable logistic regression analysis…., GSHS 2017”.

Thanks indeed for pointing this out. In the revised manuscript the title has been revised accordingly. And the table 3, 4, and 5 have been merged as per another comment. (Page 13, revised Table 3)

#20 Page 17, Table 5: Reference category for none of the variables are indicated. Reference category for each variable should be included.

All explanatory variables used are binary (1/0 coded) variables. It is respectfully pointed out that when reporting binary variables, it is the statistical norm that presence/yes/1 is compared against absence/no/0. 

#21 Page 13, 15, 17: Tables 3 to 5: Instead of using three Tables on ORs – the authors can presents results of Tables 3, 4 and 5 as one Table – so the readers can see the differences of unadjusted and adjusted ORs side by side in same Table with following format below. The new Table can be titled as

“Table 3: Unadjusted odds ratios (ORs) from univariate logistic regression analysis and adjusted ORs from multivariable logistic regression analysis with 95% confidence interval (95%CI) for the association … Paraguay, GSHS 2017 [[see the attached pdf file]

Tables 3-5 have been combined as per comment in the revised manuscript. (Page 13, revised Table 3)

Reviewer #2: Title of the manuscript: Physical fighting and associated factors among adolescents in Paraguay the 2017 national school-based health survey

Thank you for the opportunity to review this paper. The aim of this study was to investigate the prevalence and correlates of physical fighting among adolescents in Paraguay based on the 2017 national school-based health survey. The manuscript addresses an interesting topic; however, there are some issues that need to be improved prior to publication. I believe this manuscript can be considered for publication after addressing the comments below:

Title: Physical fighting and associated factors among adolescents in Paraguay the 2017 national school-based health survey.

I suggest modifying the title to “Prevalence and correlates of Physical fighting among adolescents in Paraguay: Findings from the 2017 national school-based health survey.”

We have revised the title accordingly. Thank you. (Page 1, line 2-3)

Abstract:

1. The English grammar and style should be checked throughout the manuscript.

We asked a professional English-language editor to edit the revised manuscript. We carefully checked every correction and/or suggestion and made further revisions to the manuscript.

Background: Line 28 – Causes of deaths – spelling error. Please change to causes of death.

We changed it to causes of death. (Page 2, line 31)

2. Lines 30-31 – Sentence seems unfinished. The extent to which previously identified factors ..what?

We rewrote it as “the extent to which previously identified factors correlate physical fighting.” (Page 2, line 34)

3. Lines 43-44 – Please combine as one sentence.

We combined the first and second sentences into one sentence. (Page 2, line 46-47)

Introduction:

1. The English grammar and style should be checked throughout the manuscript.

We asked a professional English-language editor to edit the revised manuscript. We carefully checked every correction and/or suggestion and made further revisions to the manuscript.

2. Lines 83-84 “tells us that” – too colloquial? Perhaps change to something formal.

We changed it to “shows.” (Page 5, line 185)

3. Lines 87 – “41% in physically attacked and 46% in physical fighting” – word missing - change to “41% in being physically attacked and 46% involved in physical fighting”

Based on another reviewer’s advice, we rewrote this part as below. (Page 5, line 189-190)

the prevalence was higher among boys than girls for physically attacked (41%, 29%) and physical fighting (46%, 27%) respectively

4. Line 102 – change to “at least one physical fight in the last 12 months”

We changed it to “at least one physical fight in the last 12 month.” (Page 6, line 296)

Materials and Methods:

1. No mention of study design used. I strongly recommend the authors to refer to STROBE checklist for cross-sectional/observational studies https://www.strobe-statement.org/

The comment has been addressed in the revised manuscript as below. (Page 6, lines 309-310) 

This methodology involves cross-sectional surveys using a self-administered questionnaire.

2. Key information such as number of schools, age of the participants included in the analysis and the eligibility criteria are missing in this section.

Thanks for pointing this out! 

Age: The age and its coding has been revised to more clearly reflect how ages were originally provided with the dataset and how we coded it for our analysis.

There were 23 records for which age was provided in the dataset as ‘less than 11 years’ and for 122 records for which age was provided as 18 years or older. This has been clarified in Table 1, in the revised manuscript. (Page 8, Table 1)

Eligibility criteria: Kindly find a new, following paragraph (Page 6, line 311-315)

GSHSs use a two-stage cluster sample design to produce representative data for all students in the specified grades in the specified year groups. The first stage entails the selection of the schools with a probability proportional to the enrolment size, and the second stage classes are randomly selected. All students present in the selected classes on the day of the survey administration are eligible to participate in the survey.

Number of schools: Each country that conducts GSHS is supposed to produce a country report detailing among other things, the total number of schools selected. However, a re-check on February 15 2022, does not show this report from Paraguay for the 2017 GSHS. However, schools are selected to represent the entire country, and GSHSs being two-stage cluster surveys, select schools in the first stage with probability proportional to enrollment size. 

Lines 122-123 – Combine both sentences.

The two sentences are combined in the revised manuscript as under: (Page 7, line 419-420)

Paraguay, 3,149 students at Octavo-Tercer Curso completed the survey questionnaire; the school, student, and total response rates as 100%, 87%, and 87%, respectively.

3. Lines 135-138 – What is the rationale behind classifying two or more fights as having participated in a physical fight and not one or more fights?

The sentence “two or more fights as having participated in a physical fight” in itself is contradictory.

It boils down to attributes of adolescent behaviour and adaptive socialization. In many ways, they are still learning how to appropriately deal with disagreement, empathy and disappointment. Thus, the occasional push or shove should not be considered problematic behaviour within the context of correlates for serious behavioural and/or mental pathology. Also, in particular for boys, within this adaptive process, some aggressivity is in many ways part of their development until they learn appropriate ways of dealing with such things as a disappointment.

What we really want to see is the truly problematic behaviour, such as involvement in physical fights repeatedly within a period of recall. 

Several published studies using GSHSs have also used involvement in 2 or more physical fights as constituting such involvement. 

4. Line 163 – Incomplete sentence. Please check grammar and sentencing throughout.

This sentence has been updated in the revised manuscript as below. (Page 9, line 597-598)

This step was followed by two additional survey binary multivariable logistic regression models.

5. Statistical analysis must talk about descriptive analysis of baseline characteristics (presented in Table 2) followed by the main analysis.

In ‘Statistical analysis’ in ‘Methods’ section it was stated as below: (Page 9, line 590-592)

“First, the distribution of selected independent variables in the entire sample was examined, followed by the distribution of selected independent variables within the dichotomised physical fights involvement variable.”

We will be grateful for more elaboration as to what additional information needs to be added here, please.

6. How was multicollinearity checked? Please elaborate with the type of analysis.

We used the survey version of Variance Inflation Factor (VIF). The VIF was considered acceptable if it was less than 10. In our final regression model (model 2 – as provided in the revised table 3) VIF was less than 3 for each explanatory variable.

Results:

1. Table 2 – Indicate number of participants along with percentages for each category and the number with missing data for each variable of interest. Please refer to the STROBE checklist.

Table 2 has been revised accordingly. (Page 11)

2. I suggest conducting Pearson’s chi-square test to check if the characteristics of those involved in physical fights were different from those who were not.

Table 2 has been revised accordingly. (Page 11)

3. Lines 206 – 12 out of 16 variables were significantly associate – not 14.

Thanks for pointing this out! In the revised manuscript, this mistake has been corrected. (Page14, line 710)

4. Line 217 – Typo – Uppercase “T” for table 3.

The tables 3 to 5 have been merged together as per the comment of the other reviewer as well, and this typo has been corrected. (Page 14, line 708)

5. Table 5 – Sex to be modified to Sex (Male).

This typo has been corrected in the revised manuscript. (Page 13)

6. Tables 2-5 – I would suggest showing both categories for each variable for clarity and readability. For example, in Table 5,

Currently, it is not a table with standalone information, we need refer back to Table 1.

Variable Adjusted OR 95% CI p-value

Anxiety 

Never/rarely/sometimes 1.00 Reference category 

Most of the time/always 1.04 0.49-2.22 0.911

The tables 3 to 5 have been merged together as per the comment of the other reviewer as well, and this typo has been corrected. (Page 13)

Since there are only three tables and in order to avoid presenting a rather crowded table, it would be easier for the reader to refer to just one table when reading the other two tables. 

Discussion:

1. Lines 252-253 – Rephrase the sentence to “Consistent with other studies,…”

We changed “In common with many studies with various methods” to “Consistent with other studies.” (Page 15, line 793)

2. Lines 274 – I would suggest using the word “association” or “odds” given a regression analysis was conducted rather than use of term “correlation”

We changed this to association. (Page 16, line 850)

We also changed the other two “correlation” to association(s) as below:

Moreover, other studies identified the associations between alcohol use, early sexual debut, and suicide planning (41, 42). (Page 16, line 858)

This study found that having been physically attacked shows the highest association with physical fighting among the variables, while the association between being bullied and physical fighting was not statistically significant in the analysis adjusted for all variables. (Page 18, line 960)

3. Lines 332-335 – Please expound on what the other sorts of interpersonal violence are based on your literature review.

We added the other sorts of interpersonal violence in this sentence which are mentioned in the Introduction section. It includes bullying, emotional abuse, sexual and physical assault, and homicide. (Page 19, line 992-994)

4. Lines 351- Please change to …accelerate actions for “them”

Based on the English-language editor’s suggestion, we rewrote this part as below: (Page 19, line 1012-1013)

We hope the study could be useful for policy makers when developing health strategies and plans for adolescents and in accelerating such actions. 

5. The findings have been critically appraised with other literature/research findings, however, the “so what” factor, that is, implications of study findings to individuals/community and policy is vague.

Thank you very much for your comments. In the Discussion session, our study highlights a couple of implications as below. To make these implications clearer, we added the summary of those points into the Conclusion section. (Page 20, line 1051-1061) 

***

1. Policy and community levels: strengthening social norms, values, and practical response

To make progress in the prevention of violence and associated factors such as alcohol use, early sexual debut, and suicide planning, Paraguay has shown better progress in the areas of implementation and enforcement of laws, caregiver support, and school-based preventions than other upper middle-income countries in the same regions. However, Paraguay lags behind in strengthening social norms and values as well as practical responses such as mental health services for both victims and perpetrators. Those areas should be strengthened. (Kindly refer to the highlighted part on Page 17, line 896-907)

2. Policy, community and family/individual levels: promoting parental support

Although the Law to prohibit all corporal punishment of children was approved and enacted in Paraguay in 2016, many families still regard parents’ corporal punishment as a teaching measure. Considering the facts that the countries with bans on corporal punishment in schools and at home have seen less physical fighting and that parental support is seen as a preventive attribute in the studies, Paraguay should also move ahead in making shifts in the relationship between parents and children by tireless awareness raising and practical executions of legislation. (Kindly refer to the highlighted part on Page 17-18, line 914-950)

3. Policy level: increasing assessment and strengthening comprehensive strategies

The Region of the Americas is likely to show a lower prevalence of problematic physical fighting among adolescents than other regions. This study found that within the region, the prevalence in Paraguay is likely to be in the average or lower groups. Considering interpersonal violence as a leading cause of deaths and DALYs in the country, other sorts of interpersonal violence including bullying, emotional abuse, sexual and physical assault, and homicide as well as their contexts should be assessed so that comprehensive strategies and measures could be introduced to protect Paraguayan adolescents from problematic violence. (Kindly refer to the highlighted part on Page 19, line 992-996)

***

The summary which was added in the Conclusion section: (Page 20, line 1051-1061)

Considering serious interpersonal violence is causing deaths among Paraguayan adolescents, other kinds of violence including bullying, emotional abuse, sexual and physical assault, and homicide, as well as their backgrounds, should be assessed in order that comprehensive strategies and measures will be introduced to protect Paraguayan adolescents from problematic violence. Furthermore, in addition to the good progress in the implementation and enforcement of laws to advance the prevention of violence, it is be critical to strengthen social norms and values against violence as well as practical responses such as providing mental health services for both victims and perpetrators. To support the prevention of violence, the bans on corporal punishment in schools and at home and the changes in the relationship between parents and children should be strongly promoted by tireless awareness raising and the practical executions of the legislation. 

Conclusion:

1. Lines 354-355 – Useful how? Please rephrase to say how the findings may be useful to inform policy and planning. What intervention programs are in place as best practice in Paraguay or other countries/ what is recommended in other countries?

We added the summary described above in the Conclusion section. (Page 20, line 1051-1061)

---

## [Decision Letter · Decision Letter 1]

8 Jun 2022

PONE-D-21-03008R1Prevalence and correlates of Physical fighting among adolescents in Paraguay: Findings from the 2017 national school-based health surveyPLOS ONE

Dear Dr. Taniguchi,

Thank you for submitting your manuscript to PLOS ONE. After careful consideration, we feel that it has merit but does not fully meet PLOS ONE’s publication criteria as it currently stands. Therefore, we invite you to submit a revised version of the manuscript that addresses the points raised during the review process.

There has been a change in the academic editor since the previous draft. The two referees were available for comments indicating that their suggested changes had been adopted. The following are the comments from the new academic editor.

There is still a need for an important copy-edit. Many sentences do not make grammatical sense or are clumsily written. I give some instances, but you should carry out a thorough revision with a native English speaker and/or grammatical software such as Grammarly.com.

Overall, it is felt that reporting is not complete, in particular regarding the model formulation. PLOS ONE endorses the use of the STROBE checklist, www.strobe-statement.org , to ensure thorough reporting.

Another general comment: The rationale of the study design is not clear: the use of the 2-fights, the selection of variables. These aspects should be improved.

Line 39-40: Remove the second “only”L. 90: Experienced physical attacks, or “were physically attacked”. Not “experienced 91 physically attacked”.Again in l. 94. In general, I’d suggest to use “physical attack” instead of “physically attacked” which is not grammarly. Eg: lowest prevalence of physical attackL. 96: Should be “the same method of analysis”Interesting that El Salvador is mentioned as having low prevalence of physical attack despite having the highest homicide rates in the world in 2019 (https://ourworldindata.org/homicides) and the highest proportion of deaths from homicides. What could be going on? Eg: One possible caveat of a retrospective study such as this is the possible lethality of the physical attacks. The survivors will have less attacks. Another possibility is underreporting due to fear of possible implications of declaring an attack. A third, sample selection of the less violent, the more violent being out of school. All of these factors could be mentioned as a limitation of this type of study.Line 114-116: You should give references and mention what they found regarding the correlates of physical fighting. The variables are coming out of the blue now and you are not explaining the rationale for their inclusion. You are mentioning some of these aspects in the discussion. This is OK. But these studies should have been mentioned and summarized in the introduction so that the reader knows what to expect and why.L. 120-128: Try to improve the writing. Eg: Avoid reiteration of "use", of "specified". L. 126, should be “in” the second stage,You do not mention the use of sampling weights in the analysis. You should use them given the 2-stage design to ensure that the analysis is statistically representative. (You mention weighting the proportions in line 186, but then table 2 is contradictorily not weighted. You also should use and mention weights in the logistic regression).Why to focus on 2 or more fights? Why not dealing with the original variable with an ordered-logit, for instance? Please submit your revised manuscript by Jul 23 2022 11:59PM. If you will need more time than this to complete your revisions, please reply to this message or contact the journal office at plosone@plos.org. Please include the following items when submitting your revised manuscript:A rebuttal letter that responds to each point raised by the academic editor and reviewer(s). You should upload this letter as a separate file labeled 'Response to Reviewers'.A marked-up copy of your manuscript that highlights changes made to the original version. You should upload this as a separate file labeled 'Revised Manuscript with Track Changes'.An unmarked version of your revised paper without tracked changes. You should upload this as a separate file labeled 'Manuscript'.If applicable, we recommend that you deposit your laboratory protocols in protocols.io to enhance the reproducibility of your results. Protocols.io assigns your protocol its own identifier (DOI) so that it can be cited independently in the future. For instructions see: https://journals.plos.org/plosone/s/submission-guidelines#loc-laboratory-protocols. Additionally, PLOS ONE offers an option for publishing peer-reviewed Lab Protocol articles, which describe protocols hosted on protocols.io. Read more information on sharing protocols at https://plos.org/protocols?utm_medium=editorial-email&utm_source=authorletters&utm_campaign=protocols.

We look forward to receiving your revised manuscript.

Kind regards,

José Antonio Ortega, Ph.D.

Academic Editor

PLOS ONE

Journal Requirements:

Reviewers' comments:

Reviewer's Responses to Questions

**Comments to the Author**

1. If the authors have adequately addressed your comments raised in a previous round of review and you feel that this manuscript is now acceptable for publication, you may indicate that here to bypass the “Comments to the Author” section, enter your conflict of interest statement in the “Confidential to Editor” section, and submit your "Accept" recommendation.

Reviewer #1: All comments have been addressed

Reviewer #2: All comments have been addressed

2. Is the manuscript technically sound, and do the data support the conclusions?

Reviewer #1: Yes

Reviewer #2: Yes

3. Has the statistical analysis been performed appropriately and rigorously? 

Reviewer #1: Yes

Reviewer #2: Yes

4. Have the authors made all data underlying the findings in their manuscript fully available?

Reviewer #1: Yes

Reviewer #2: Yes

5. Is the manuscript presented in an intelligible fashion and written in standard English?

Reviewer #1: Yes

Reviewer #2: Yes

6. Review Comments to the Author

Reviewer #1: The authors addressed issues raised on original submitted version. Specifically methods section, Correction of typos in Tables and combining Tables 3-5 as one Table. Due to revision the quality of the article has improved.

Reviewer #2: The authors have addressed all the comments which has made improvements in the revised manuscript. No further changes.

7. PLOS authors have the option to publish the peer review history of their article (what does this mean?). If published, this will include your full peer review and any attached files.

Reviewer #1: No

Reviewer #2: **Yes: **James Rufus John

---

## [Author Response · Author response to Decision Letter 1]

26 Aug 2022

Manuscript ID: PONE-D-21-03008

Response to Reviewers

To

Dr José Antonio Ortega

Academic Editor

PLOS ONE

Dear Dr José Antonio Ortega

Thank you very much for your consideration of our manuscript, Physical fighting and associated factors among adolescents in Paraguay the 2017 national school-based health survey. We have revised the manuscript thoroughly according to the comments by the editor. We addressed all points clearly in the revised manuscript.

Our responses to the editor are provided as below. Kindly find our response each point.

Also, we have revised the manuscript and the corresponding sentences in the texts have been highlighted using track changes. All line numbers referred to in our responses below are in the margins of our revised manuscript with tracked changes. 

We hope that the revisions are satisfactory in addressing issues raised by the editor and look forward to hearing your decision about this article.

Yours sincerely,

Authors

There has been a change in the academic editor since the previous draft. The two referees were available for comments indicating that their suggested changes had been adopted.

->Thank you very much for sharing these.

There is still a need for an important copy-edit. Many sentences do not make grammatical sense or are clumsily written. I give some instances, but you should carry out a thorough revision with a native English speaker and/or grammatical software such as Grammarly.com.

->Thank you for your suggestion. The manuscript was re-edited by an English-language editor and, in addition, was double-checked and updated using Grammarly.com.

Overall, it is felt that reporting is not complete, in particular regarding the model formulation. PLOS ONE endorses the use of the STROBE checklist, www.strobe-statement.org, to ensure thorough reporting.

->We filled in the STROBE checklist and will include it in the submission. Could you find it for your reference?

The rationale of the study design is not clear: the use of the 2-fights.

- Why to focus on 2 or more fights? Why not dealing with the original variable with an ordered-logit, for instance?

->We classified fighting behaviour into what could be considered potentially problematic (which has the possibility to result in short or long term sequalae) and potentially non-problematic (rough play). This categorization has been taken into consideration and now well understood from the psychosocial aspects of adolescent development and behaviour. For example, rough play should not in and of itself be classified alongside problematic behavioural patterns. It is instead characteristic of rapid bio-psychological, psycho-social and environmental adjustment as adolescents navigate their way into adulthood. 

The aim with the above classification lies in avoiding pathologizing adolescent development patterns, particularly the behaviour of adolescent males who may display higher levels of physical aggression, but this aggression does not in and of itself constitute problematic behaviour.

Studies on adolescent aggression demonstrate that there is a difference between aggression and roughness among adolescents. Roughness during play may be associated with aspects of peer affiliation, although those on the other side of the aggression may need to defend themselves (1). Rough play is when a fight occurs without demanding submissiveness and distress to the victim (1). On the other hand, aggressiveness goes beyond roughness when the aggressor is likely to persist until they achieve a level of submissiveness from the victim (2). In this regard, the intention is to achieve dominance over peers and is likely to target potential victims who appear weaker in order to defeat them, and achieve a higher dominance reflected by the number of victories (2). Engagement in roughness and escalation to aggression with other peers is done to demonstrate dominance by taking on various fights (2).

In relation to the present study, the justification for cut-off in fights occurring at least 2 times or more was to include adolescents who may have been involved in problematic aggressive behaviour. Thus, the problematic behaviour is when one person is considered to have been involved in fights several times, either as the aggressor or the individual on the receiving end of the fighting behaviour. The non-problematic behaviour was considered to have occurred when one was not involved in fights at all or only once during the 12 months preceding the survey. This is because participation in one fight may not have been with the intention to achieve dominance but rather the fight could have been a random occurrence or a one-off disagreement with a peer. The challenge is that data on the reason for participating in the fights was not available and setting the cut-off at this level was to distinguish between frequent fighting and no or non-frequent fighting.

We described the point in the manuscript. (Line 187-192)

The rationale of the study design is not clear: the selection of variables.

->The variables were selected based on previous studies which studied the physical fighting among school-attending adolescents using GSHS data from various other countries as cited in reference numbers (3, 11-16, 22-26). (Line 169-170)

The choice of variables in the final logistic regression model were based on the results of bivariate analyses. (Line 226-228)

- Line 39-40: Remove the second “only”

->Thank you. We deleted it. (Line 39)

- L. 90: Experienced physical attacks, or “were physically attacked”. Not “experienced 91 physically attacked”.

->We discussed and revised as “having been physically attacked” to keep the interviewees’ passive experiences in the past. (Line 99)

- Again in l. 94. In general, I’d suggest to use “physical attack” instead of “physically attacked” which is not grammarly. Eg: lowest prevalence of physical attack

->We discussed and revised as “having been physically attacked” to keep the interviewees’ passive experiences in the past. (Line 100-101)

- L. 96: Should be “the same method of analysis”

->We revised it. (Line 104-105)

- Line 114-116: You should give references and mention what they found regarding the correlates of physical fighting. The variables are coming out of the blue now and you are not explaining the rationale for their inclusion. You are mentioning some of these aspects in the discussion. This is OK. But these studies should have been mentioned and summarized in the introduction so that the reader knows what to expect and why.

->Thank you for your suggestion. We added the summarised findings from the prior studies regarding the association between physical fighting among school-attending adolescents and family- and peer-level factors. (Line 107-109)

- L. 120-128: Try to improve the writing. Eg: Avoid reiteration of "use", of "specified". 

->We revised the paragraph (Line 139-140) as well as the others in the whole manuscript. 

- L. 126, should be “in” the second stage,

->We revised it as suggested. (Line 141)

- You do not mention the use of sampling weights in the analysis. You should use them given the 2-stage design to ensure that the analysis is statistically representative. (You mention weighting the proportions in line 186, but then table 2 is contradictorily not weighted. You also should use and mention weights in the logistic regression).

->The last paragraph/sentence in the ‘Statistical analysis’ section is revised to clarify the analysis approach used, as under: (Line 232-235)

“Design-based analyses were carried out using Stata 16 program (StataCorp, 2019), by taking into account the complex survey design. All proportions, results of chi-square tests, and logistic regression models are reported based on design-based analysis; while unweighted counts/frequencies are reported.”

In table 2, the square brackets show the unweighted counts i.e. frequencies, while all the reported percentages are weighted; the results of tests and their p-values are all based on the statistical tests which accounted for the complex survey design. 

Table 3 shows the results of design-based logistic regression models which took into account the complex survey design.

We also added these points to the tables. (Line 264-267; Line 291)

- Interesting that El Salvador is mentioned as having low prevalence of physical attack despite having the highest homicide rates in the world in 2019 (https://ourworldindata.org/homicides) and the highest proportion of deaths from homicides. What could be going on? Eg: One possible caveat of a retrospective study such as this is the possible lethality of the physical attacks. The survivors will have less attacks. Another possibility is underreporting due to fear of possible implications of declaring an attack. A third, sample selection of the less violent, the more violent being out of school. All of these factors could be mentioned as a limitation of this type of study.

->These are indeed important points. We added two points in the limitation. 

Students who were physically attacked to the point of death were not included in the survey (survivorship bias). (Line )

Third, this study used a self-report survey. This might have introduced a social-desirability bias and a relevant bias to the study results. It includes fears of possible implications of declaring attacks. (Line 459-460, 464-465)

Reference

1. Neill SRSJ. AGGRESSIVE AND NON-AGGRESSIVE FIGHTING IN TWELVE-TO-THIRTEEN YEAR OLD PRE-ADOLESCENT BOYS. Journal of Child Psychology and Psychiatry. 1976;17(3):213-20.

2. Pellegrini AD. A longitudinal study of boys' rough-and-tumble play and dominance during early adolescence. Journal of Applied Developmental Psychology. 1995;16(1):77-93.

---

## [Editor Report · Decision Letter 2]

22 Sep 2022

PONE-D-21-03008R2Prevalence and correlates of Physical fighting among adolescents in Paraguay: Findings from the 2017 national school-based health survey

PLOS ONE

Dear Dr. Taniguchi,

Thank you for submitting your manuscript to PLOS ONE. After careful consideration, we feel that it has merit but does not fully meet PLOS ONE’s publication criteria as it currently stands. Therefore, we invite you to submit a revised version of the manuscript that addresses the points raised during the review process.

It is felt that the manuscript has substantially improved. A few minor changes, mostly typos, emerge from the reading.

l. 101 Should be: were regarded **as** protective attributes

Both in the abstract and in the main text you mention “3,149 students at Octavo-Tercer Curso completed the survey questionnaire”. This is not clearly understandable. It literally means in Spanish “eight-third course”. I assume it is eight year of primary to third year of secondary, but this is unclear. Why not refer in both instances, as described in the survey documentation “UNIVERSE School-going adolescents aged 13-17 years.”

Line 145: You describe variable by variable missing information. It would be good to add a summary sentence saying: As a result a total of ---- observation were dropped due to missing information on some variable”. Note that several variables can be missing. If my reading is incorrect and each sentence refers to additional missing cases, then rephrase and make it clear.

L. 264: should be “the two most protective associations were the attributes of having helpful peers and supportive parents”

L. 270: Counties should be countries

L 361: Multicountry, not multicounty

L. 378: to the point of death or dropout might be prefarrable. You are also missing children not at school.

We look forward to receiving your revised manuscript.

Kind regards,

José Antonio Ortega, Ph.D.

Academic Editor

PLOS ONE
---

## [Author Response · Author response to Decision Letter 2]

1 Nov 2022

Manuscript ID: PONE-D-21-03008

Response to Reviewers

To

Dr José Antonio Ortega

Academic Editor

PLOS ONE

Dear Dr José Antonio Ortega

Thank you very much for your consideration of our manuscript, Physical fighting and associated factors among adolescents in Paraguay the 2017 national school-based health survey. We have revised the manuscript thoroughly according to your comments. We addressed all points clearly in the revised manuscript.

Our responses are provided as below. Kindly find our response in blue per each point.

Also, we have revised the manuscript and the corresponding sentences in the texts have been highlighted using track changes. All line numbers referred to in our responses below are in the margins of our revised manuscript with tracked changes. 

We hope that the revisions are satisfactory in addressing issues and look forward to hearing your decision about this article.

Yours sincerely,

Authors

Thank you for submitting your manuscript to PLOS ONE. After careful consideration, we feel that it has merit but does not fully meet PLOS ONE’s publication criteria as it currently stands. Therefore, we invite you to submit a revised version of the manuscript that addresses the points raised during the review process.

It is felt that the manuscript has substantially improved. A few minor changes, mostly typos, emerge from the reading.

Thank you very much for your comments and careful review.

l. 101 Should be: were regarded as protective attributes

We revised the part as suggested. (Line 102)

Both in the abstract and in the main text you mention “3,149 students at Octavo-Tercer Curso completed the survey questionnaire”. This is not clearly understandable. It literally means in Spanish “eight-third course”. I assume it is eight year of primary to third year of secondary, but this is unclear. Why not refer in both instances, as described in the survey documentation “UNIVERSE School-going adolescents aged 13-17 years.”

In Abstract, the sentence has been revised as below:

This survey collects health-related information on school-attending adolescents aged 13-17 years. (Line 35)

And in Materials and Methods section as below:

In Paraguay, 3,149 students at the age of 13-17 years completed the survey questionnaire

(Line 138)

Line 145: You describe variable by variable missing information. It would be good to add a summary sentence saying: As a result a total of ---- observation were dropped due to missing information on some variable”. Note that several variables can be missing. If my reading is incorrect and each sentence refers to additional missing cases, then rephrase and make it clear.

Thank you for this comment. Owing to missing information in various covariates, the total number of observations/records used in the final multivariable logistic model were 2,520. We added the following sentence to further clarify.

“As a result, a total of 2,520 observations were used in the final multivariable logistic regression model, for information was available for all variables.” (Line 253-254)

L. 264: should be “the two most protective associations were the attributes of having helpful peers and supportive parents”

We revised the part as suggested. (Line 269-273)

L. 270: Counties should be countries

We revised the part as suggested. (Line 278)

L 361: Multicountry, not multicounty

We revised the part as suggested. (Line 369)

L. 378: to the point of death or dropout might be prefarrable. You are also missing children not at school.

Thank you for your suggestion. We agreed on it and added dropout in the sentence as underlined.

Regarding children not at school, we described children who are not at school due to various reason in the prior description and added another example as underlined. 

“First, this is a cross-sectional study which captured only the responses of the students who attended school on the day of the survey. It could mean, for example, that although our study did not identify food deprivation as a prevalent attribute, students from impoverished households might not have been at school on that day or might not have been attending school at all. Students with only a few close friends might also have been unwilling to be at school. Students who tend to play truant from schools may not be included in the survey. Students who were physically attacked to the point of death or dropout were not included in the survey (survivorship bias).” (Line 380-387)

Note:

For the blue-colored texts and the underlined texts, could you check blue and underlines in the attached "Response to Reviewers" file?

---

## [Editor Report · Decision Letter 3]

15 Nov 2022

PONE-D-21-03008R3Prevalence and correlates of Physical fighting among adolescents in Paraguay: Findings from the 2017 national school-based health surveyPLOS ONE

Dear Dr. Taniguchi,

Thank you for submitting your manuscript to PLOS ONE. After careful consideration, we feel that it has merit but does not fully meet PLOS ONE’s publication criteria as it currently stands. Therefore, we invite you to submit a revised version of the manuscript that addresses the points raised during the review process.

The article is basically ready for publication, except for the quality of the English that still requires improvement. In particular the new edited paragraphs do not read well. Also, maybe you are not interpreting correctly the concern on not covering the complete population. It is a concern only to the extent that this is implies selection on your dependent variable, physical fighting. Also some judgmental expressions like playing truant are better to be avoided. Also you acknowledged that the design does not allow causal statements, but the choice of variables also does not help. Eg: could not attacks be a result of previous fights and not otherwise?

Please replace

First, this is a cross-sectional study which captured only the 375 responses of the students who attended school on the day of the survey. It could mean, for 376 example, that although our study did not identify food deprivation as a prevalent attribute, 377 students from impoverished households might not have been at school on that day or might 378 not have been attending school at all. Students with only a few close friends might also have 379 been unwilling to be at school. Students who tend to play truant from schools may not be 380 included in the survey. Students who were physically attacked to the point of death or dropout 381 were not included in the survey (survivorship bias). Second, cross-sectional data do not confer 382 causal relationships between attributes studied.

With

First, this is a cross-sectional study which captured only the responses of the students who attended school on the day of the survey. School attendance could be correlated with the risk of physical fighting and being physically attacked due to sequels of previous episodes or preventive behaviour. Second, the cross-sectional nature of the data does not allow causal interpretation of the associations studied. Reverse causality could also affect behavioural variables such as being physically attacked.

389 among school-attending adolescents in Paraguay. We also used the most** updated** survey data

Change updated to **recent**

We look forward to receiving your revised manuscript.

Kind regards,

José Antonio Ortega, Ph.D.

Academic Editor

PLOS ONE
---

## [Author Response · Author response to Decision Letter 3]

5 Dec 2022

Manuscript ID: PONE-D-21-03008

Response to Reviewers

To

Dr José Antonio Ortega

Academic Editor

PLOS ONE

Dear Dr José Antonio Ortega

Thank you very much for your consideration of our manuscript, Physical fighting and associated factors among adolescents in Paraguay the 2017 national school-based health survey. We have revised the manuscript thoroughly according to your comments. We addressed all points clearly in the revised manuscript.

Our responses are provided as below. Kindly find our response per each point.

Also, we have revised the manuscript and the corresponding sentences in the texts have been highlighted using track changes. All line numbers referred to in our responses below are in the margins of our revised manuscript with tracked changes. 

We hope that the revisions are satisfactory in addressing issues and look forward to hearing your decision about this article.

Yours sincerely,

Authors

Thank you for submitting your manuscript to PLOS ONE. After careful consideration, we feel that it has merit but does not fully meet PLOS ONE’s publication criteria as it currently stands. Therefore, we invite you to submit a revised version of the manuscript that addresses the points raised during the review process.

->Thank you very much for your comments and careful review.

The article is basically ready for publication, except for the quality of the English that still requires improvement. In particular the new edited paragraphs do not read well. Also, maybe you are not interpreting correctly the concern on not covering the complete population. It is a concern only to the extent that this is implies selection on your dependent variable, physical fighting. Also some judgmental expressions like playing truant are better to be avoided. Also you acknowledged that the design does not allow causal statements, but the choice of variables also does not help. Eg: could not attacks be a result of previous fights and not otherwise?

Please replace

First, this is a cross-sectional study which captured only the 375 responses of the students who attended school on the day of the survey. It could mean, for 376 example, that although our study did not identify food deprivation as a prevalent attribute, 377 students from impoverished households might not have been at school on that day or might 378 not have been attending school at all. Students with only a few close friends might also have 379 been unwilling to be at school. Students who tend to play truant from schools may not be 380 included in the survey. Students who were physically attacked to the point of death or dropout 381 were not included in the survey (survivorship bias). Second, cross-sectional data do not confer 382 causal relationships between attributes studied.

With

First, this is a cross-sectional study which captured only the responses of the students who attended school on the day of the survey. School attendance could be correlated with the risk of physical fighting and being physically attacked due to sequels of previous episodes or preventive behaviour. Second, the cross-sectional nature of the data does not allow causal interpretation of the associations studied. Reverse causality could also affect behavioural variables such as being physically attacked.

389 among school-attending adolescents in Paraguay. We also used the most updated survey data

Change updated to recent

->Thank you very much for your clarification and suggestions. We revised the parts as suggested. (Line 379-384, 405)

->We also had another English check through the manuscript.

END

---

## [Editor Report · Decision Letter 4]

7 Dec 2022

Prevalence and correlates of physical fighting among adolescents in Paraguay: Findings from the 2017 national school-based health survey

PONE-D-21-03008R4

Dear Dr. Taniguchi,

We’re pleased to inform you that your manuscript has been judged scientifically suitable for publication and will be formally accepted for publication once it meets all outstanding technical requirements.

Kind regards,

José Antonio Ortega, Ph.D.

Academic Editor

PLOS ONE

Additional Editor Comments (optional):

The paper is ready for publication from an academic perspective. Congratulations! Regarding language issues, you might be contacted by the journal on that respect if it is deemed necessary.
---

## [Editor Report · Acceptance letter]

19 Dec 2022

PONE-D-21-03008R4 

Prevalence and correlates of physical fighting among adolescents in Paraguay: Findings from the 2017 national school-based health survey 

Dear Dr. Taniguchi:

I'm pleased to inform you that your manuscript has been deemed suitable for publication in PLOS ONE. Congratulations! Your manuscript is now with our production department. 

Kind regards, 

on behalf of

Dr. José Antonio Ortega 

Academic Editor

PLOS ONE